



# Early results and Validation of SAGE III-ISS Ozone Profile Measurements from Onboard the International Space Station

Michael P. McCormick, Liqiao Lei, Michael T. Hill

Center for Atmospheric Sciences, Department of Atmospheric and Planetary Sciences, Hampton University, Hampton, VA, 23668, USA

*Corresponding to*: M. Patrick McCormick (pat.mccormick@hamptonu.edu)

**Abstract.** The Stratospheric Aerosol and Gas Experiment III (SAGE III) instrument, was launched on February 19, 2017 from the NASA Kennedy Space Center, and integrated aboard the International Space Station (ISS). SAGE III-ISS has been providing ozone profile measurements since June, 2017. This paper presents an early validation of the Level 2 solar and lunar occultation ozone data products using ground-based lidar and ozonesondes from Hohenpeissenberg and Lauder, and satellite ozone vertical products from the Atmospheric Chemistry Experiment Fourier-Transform Spectrometer (ACE-FTS) instrument. The Hohenpeissenberg one-year lidar results show that the average difference of ozone concentration measured by SAGE III-ISS is less than 10% between 15 and 45 km and less than 5% between 20 and 40 km. Hohenpeissenberg ozonesonde comparisons are mostly within 10% between 15 and 30 km. The Lauder lidar comparison results were less than 10% between 17 and 40 km, and less than 10% between 10 km and 30 km for Lauder ozonesondes. The seasonal average difference of ozone concentration between SAGE III-ISS and ACE-FTS was mostly less than 5% between 20 and 45 km for both the northern and southern hemispheres. All results from these comparisons show that the SAGE III-ISS ozone solar data have exceptional accuracy between 20 and 30 km, and believable accuracy throughout the stratosphere. With few comparisons available, the percentage difference between the SAGE III-ISS lunar ozone data and the ozonesonde data is less than 10% between 20 and 30 km. The percentage difference between the SAGE III-ISS lunar ozone data and the ACE-FTS ozone data is less than 10% between 20 and 40 km.

## 1. Introduction

Ozone plays a significant role in the atmosphere because it contributes to the radiative balance of the atmosphere by absorbing ultraviolet (UV) solar radiation, and, in addition affects the health of humans, animals, and plants (Solomon, 1999). Therefore, it is important to understand its global variations and trends (McCormick et al., 1992; Reinsel et al., 2002; Bourassa et al., 2014; Harris et al., 2015), and its impact on climate change (Rex et al, 2004; Son et al., 2008; Thompson et al, 2011). Ozone measurements from the SAGE series of satellite instruments, including SAGE I, II, III/METEOR-3M, and III/ISS, provide important data to investigate stratospheric change and long-term variability in the vertical distribution of stratospheric ozone. The occultation technique that the SAGE series utilizes provides a consistent methodology and fundamental assumption for processing data (McCormick et al., 1989; Wang et al., 2006; Damadeo et al., 2013). The solar occultation method makes SAGE one of the best series of satellite instruments for precise and accurate stratospheric ozone measurements. Other than the SAGE series, previous solar occultation satellite instruments include the HALogen Occultation Experiment (HALOE) (Russell et al., 1993), Polar Ozone and Aerosol Measurement III (Lucke et al., 1999), and Atmospheric Chemistry Experiment Fourier Transform Spectrometer (ACE-FTS) (Bernath et al., 2005). The Scanning Imaging Absorption spectrometer for atmospheric ChartographY (SCIAMACHY) also has an occultation mode (Bovensmann et al., 1999). The ACE-FTS is still in operation and is providing valuable data for the SAGE III-ISS validation as will be shown in this paper. It is important for the SAGE III-ISS ozone profiles to be well validated to extend the long-standing ozone record of observation from the SAGE series, POAM III, HALOE, and ACE-FTS. The well characterized ozone data will contribute to the investigation of any trend and



possible ozone recovery due to a CFC reduction. In this paper, the global SAGE III-ISS ozone profile data are compared with correlative datasets to investigate possible differences. These comparisons will begin the process

of showing that SAGE III-ISS ozone data can be used for scientific studies. A systematic assessment of the early SAGE III-ISS ozone data is made by comparing SAGE III-ISS ozone profiles with ozone profiles made by Hohenpeissenberg and Lauder ground-based lidar and ozonesondes, as well as satellite data from the ACE-FTS. Section 2 describes the instrument and ozone product for the comparison, as well as the criteria for coincidence and the methodology for validation. The comparison results between coincident events are shown in Sect. 3. The

overall summary and conclusion are shown in Sect. 4.

## 2. Instruments and Methods

### 2.1 SAGE III-ISS

The SAGE III-ISS payload was launched by the SpaceX Falcon rocket on 19 February 2017 from the NASA Kennedy Space center and delivered to the ISS by the SpaceX Dragon spacecraft.   It was mated to the ISS on 7

March 2017. The ISS is flying in a 51.64° inclination low-Earth orbit, which provides low- and mid-latitude occultation coverage. Figure 1 shows SAGE III-ISS solar and lunar occultation coverage from June, 2017 to November, 2018. The primary objective of the SAGE III-ISS mission is to obtain vertical profiles of ozone, water vapor, nitrogen dioxide, nitrogen trioxide, and aerosol extinction at multiple wavelengths, using solar and lunar occultation measurements. Similar to the SAGE III/METEOR-3M,SAGE III-ISS uses an 809×10 pixel Charged

Couple Device (CCD) array to provide continuous spectral coverage from 280-1040 nm with a spectral resolution of 1 to 2 nm. Additionally, an InGaAs infrared (IR) photodiode centered at 1550 nm is included for aerosol extinction measurements at a longer wavelength (Wang et al., 2006).   Only 87 pixel groups are transmitted from the satellite for gaseous species and aerosol retrieval because of the limitation in the telemetry bandwidth. As well as solar occultation, SAGE III-ISS is capable of making lunar occultation measurements at nighttime for ozone,

nitrogen dioxide, nitrogen trioxide, and chlorine dioxide. SAGE III-ISS level 2 solar species retrievals include four retrieved ozone products: MesO3, ChapO3, Ozone_aO3, and CompositeO3.

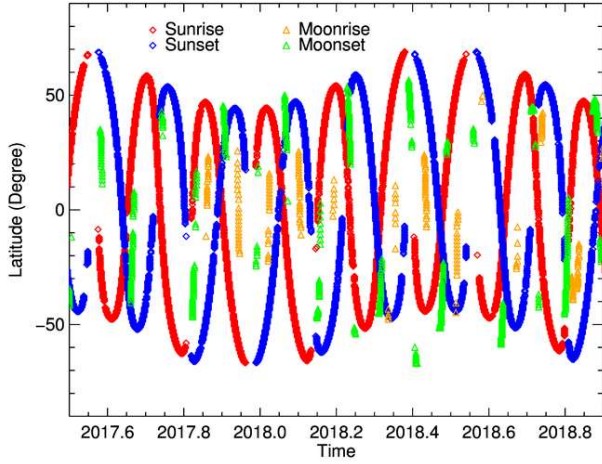

**Figure 1. The SAGE III-ISS solar and lunar occultation coverage from June, 2017 to November, 2018. The red curves show the sunrise events; the blue curves show the sunset events; the orange curves show the moon rise events and the green curves show the moonset events.**


The profile based upon the measurement made in the Hartley-Huggins band is denoted as Mesospheric ozone (MesO3). The profile based upon measurements in the Chappuis band is denoted as Multiple Linear Regression Ozone (ChapO3). The profile obtained using a similar approach to that used by SAGE II is denoted as Least

Squares Ozone (Ozone_aO3) (SAGE III-ISS Data Products User's Guide, 2017).

### 2.2 Lidar and Ozonesonde

Lidar and ozonesonde ozone profiles provided by the Network for the Detection of Atmospheric Composition Change (NDACC) were used for SAGE III-ISS ozone profile comparisons. The Hohenpeissenberg (48° N, 11° E) ozone lidar has been providing ozone profile data from 15 to 50 km since 1987.    The Hohenpeissenberg balloon

ozonesonde has been providing ozone profile data since 1967. A remote-sensing research station located at Lauder, New Zealand (45° S, 169.7° E) is also providing lidar ozone profile data from 8 to 50 km and ozonesonde ozone profiles data from 0 to 32 km since 1986.

### 2.3 ACE-FTS

The Canadian Atmospheric Chemistry Experiment (ACE) on the SCISAT-1 satellite was launched on 12 August

2003 (Bernath et al., 2005) and is operational at the time of this writing. The ACE-FTS is one of the two instruments on-board the spacecraft and provides vertical profiles of ozone and trace gases, as well as temperature, pressure, and aerosol extinction (Boone et al., 2005, Waymark et al., 2013).    ACE-FTS makes its solar occultation measurements in the 85°S to 85°N latitude region due to its circular 650 km, 74° inclination, low-Earth orbit (Bernath et al., 2005). The ACE-FTS vertical measurement range typically extends from 10-95 km for

ozone. The maximum vertical resolution of ACE-FTS is 3-4 km based on its instrument field-of-view (Dupuy et al., 2009). ACE-FTS level 2 version 3.5/3.6 data were used for the ozone comparisons in this work.

### 2.4 The Methodology for Comparisons

As a satellite with near-global coverage, SAGE III-ISS is expected to find many coincidences with correlative instruments to get a robust conclusion for the accuracy of its data (Imai, 2013). Coincidence for the ozone

measurements from SAGE III-ISS and correlative measurements are selected as the pair of profiles that has the closest geographic distance and closest time between each set of measurements. Coincidence criteria varies for different validation studies to gets sufficient amount of coincident events in all comparisons. For coincidence events between SAGE III-ISS and lidar/ozonesonde data, the criteria of less than ±5° in latitude, less than ±5° in longitude, and less than 48 hours in time were used. The coincidence events for SAGE III-ISS and ACE-FTS in

the northern hemisphere has the criteria of less than ±5° in latitude, less than ±10° in longitude, and less than ±4 hours in time. In order to get sufficient comparisons in the southern hemisphere the criteria were expanded to less than ±10° in latitude, less than ±20° in longitude, and less than ±10 hours in time.    The broad criteria could result in multiple coincidences for a single SAGE III-ISS profile. In the case of multiple matches, the coincident pair that has the smallest time and spatial difference was chosen. This process reduces the duplicate coincidence events

in the comparisons. The coincident profiles for the two correlative instruments will be found, and then the difference between the two coincident profiles are calculated after the data were screened. For statistical analysis, coincident data for the comparison were screened to reject the profiles with low-quality measurements according to the recommendation provided by each data product's user guide.    Therefore, part of the data record will be removed and this will decrease the total amount of coincident pairs in our comparisons. (Hubert, D., et al., 2016).

The coincident pair of ozone profiles from the two instruments are linearly interpolated to the SAGE III-ISS altitude grid (Rong et al., 2009; Dupuy et al., 2009).





The average difference between the coincident pairs of profiles at a given altitude are calculated using the Eq. (1):

$$\Delta(z) = \frac{1}{N(z)} \sum_{i=1}^{N} \left[ \frac{SAGEIII_i(z) - corr_i(z)}{ref_i(z)} \right] \tag{1}$$

where $\Delta(z)$ refers to the average ozone difference at a given altitude z, $SAGEIII_i(z)$ is the ozone concentration at altitude z for the i'th coincident SAGE III-ISS profile, and $corr_i$ is the corresponding concentration for the correlated comparison instrument for the i'th coincident pair. $N(z)$ is the total number of the coincident measurement pairs at altitude z, and $ref_i(z)$ is the i'th reference at altitude z for calculating the difference. The reference for calculating the absolute difference between the coincident pair equals 1. In the case of calculating the relative difference for each coincident pair, the $ref_i(z)$ is the average of the i'th coincident pair concentration at altitude z (Randall et al., 2003; Smith et al., 2013).

$$ref_i(z) = \left( SAGEIII_i + corr_i(z) \right)/2 \tag{2}$$

As shown in Eq. (3), the standard deviation of the distribution of the relative difference at altitude z provides the spread in the difference for individual coincident pairs. This provides information of the significance of the bias of the SAGEIII-ISS instrument. This standard deviation could also provide a measure of the combined precision of the instruments that were used for the comparison (von Clarmann, 2006)

$$\sigma(z) = \sqrt{\frac{1}{N(z)-1} \sum_{i=1}^{N(z)} \left[ \left( SAGEIII_i(z) - corr_i(z) - \Delta(z) \right) \right]^2} \tag{3}$$

where $N(z)$ is the total number of coincidence pairs at altitude z, $SAGEIII_i(z)$ is the concentration at altitude z for SAGE III-ISS, $corr_i$ is the corresponding concentration of the correlated comparison instrument for the i'th coincident pair, and $\Delta(z)$ refers to the average difference at a given altitude z. The statistical uncertainty of the mean difference, also known as standard error of the mean (SEM), is the quantity that allows the significance of the estimated biases to be judged (Dupuy et al., 2009).

$$SEM(z) = \sigma(z)/\sqrt{N(z)} \tag{4}$$

## 3. Results

### 3.1 Comparison between SAGE III-ISS and Ground-based Lidar

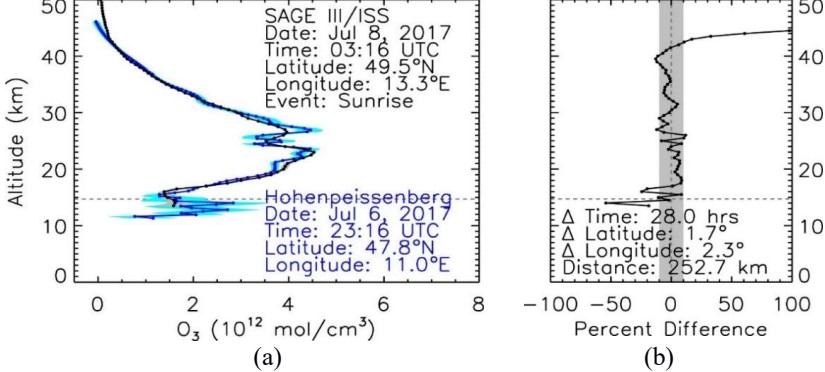

Figure 2. (a) The coincident pair of profiles for SAGE III-ISS and Hohenpeissenberg ozone lidar on July 8, 2017. The SAGE III-ISS ozone profile is shown as the black line with the precision shown as grey shading. The Hohenpeissengburg lidar ozone profile is shown as the blue line with the precision shown with cyan shading. (b) Percentage difference between the coincident pair of ozone profiles. Black line indicates the percentage difference; the vertical grey shading shows the ±10% difference region. The horizontal dash lines in (a) and (b) show the tropopause height as reported in the SAGE III-ISS data product.





135     Figure 2 shows a comparison between a SAGE III-ISS ozone profile from July 8, 2017 and a coincident profile
        from the Hohenpeissenberg lidar. The difference for this coincident event is 1.7° in latitude, 2.3 ° in longitude,
        28 hours in time, and 252.7 km in distance. The comparison shows good agreement between 17 km and almost
        40 km with the percentage difference less than 10% with few exceptions. As expected, the lidar data show high
        variability at and below the tropopause resulting in larger differences with SAGE III-ISS; a SAGE III measure-
ment that can be affected by clouds over its horizontal limb measurement.

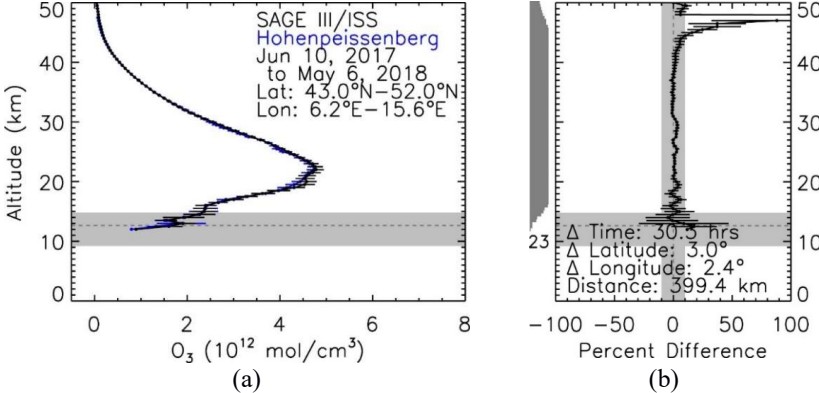

(a)                                                      (b)

**Figure 3(a). The average of all coincident pairs for SAGE III-ISS data and Hohenpeissenberg ozone lidar data for the year June 2017 to May 2018. The average SAGE III-ISS ozone profile is shown as the black line with the average precision as black error bars. The average Hohenpeissengburg lidar ozone profile is shown as a blue line with the precision shown as horizontal blue error bars. (b) The percentage difference between the average coincident pairs is shown. The black line indicates the percentage difference, the black horizontal error bars show the standard error of the difference, and the vertical grey shading shows the ±10% region. The horizontal grey shading shows the altitude range of the tropopause for both (a) and (b). The horizontal light grey dotted line shows the average altitude of the tropopause height as reported in the SAGE III-ISS data product. The vertical shading on the left side of (b) indicates the variation in the number of coincidences at each altitude (23 total).**

        One year of data, from June 2017 to May 2018, using the 23 coincident profile pairs between the SAGE III-ISS
and the Hohenpeissenberg ozone lidar ozone data, is shown in Figure 3a. The difference of all coincident pairs for
        one year of data was calculated and is shown in Figure 3b. The average time difference is 30.3 hours; average
        latitude difference is 3.0°; average longitude difference is 2.4°; and the average profile distance is 399.4 km. The
        average percentage difference between the SAGE III-ISS and the Hohenpeissenburg lidar is less than 5% from 20
        to 40 km, and 10% from 15 to 45 km with the very low standard error between 17 km and 40 km, and increasing
below and above this region.

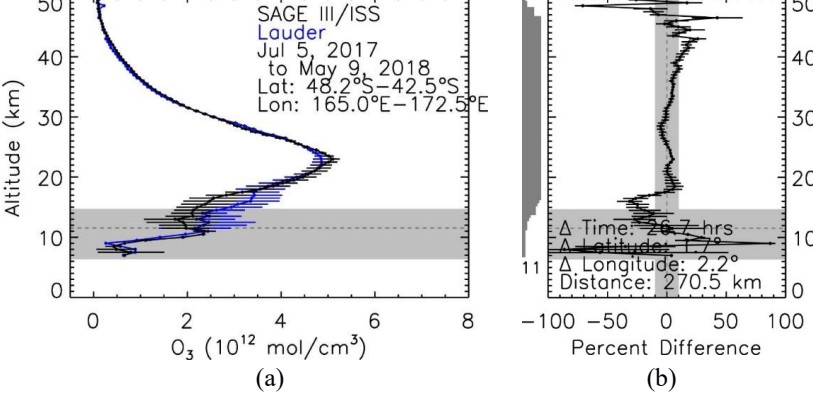

(a)                                                      (b)

**Figure 4. Similar to Figure 3, but for an average difference of coincident pairs of ozone profile differences for SAGE III-ISS and Lauder ozone lidar data for the year July 2017 through May 2018.**





Similarly, comparisons between the coincident pairs of the SAGE III-ISS and Lauder ozone lidar are shown in
figure 4. There is a total of 11 coincident pairs found within 26.7 hours average time, 1.7° average latitude
difference, 2.2° average longitude difference, and 270.5 km average separation distance. The average percentage
difference is less than 10% from about 17 to about 40 km with a low standard error between 20 to 40 km.

### 3.2 Comparison between SAGE III-ISS and Ozonesondes

The average of 29 coincident ozone profile pairs between SAGE III-ISS and Hohenpeissenburg ozonesonde was
compared as shown in Figure 5. Note that the altitude spread of these comparisons are shown as the vertical grey
bar. The coincident pairs had an average time difference of 18 hours, latitude difference of 2.8°, longitude
difference of 2.6°, and a 394.2 km distance. The average ozone percentage differences were found to be less than
10% from about 15 to 30 km. The SAGE III-ISS average ozone values show a positive difference increase above
27 km which could be a result of decreased pump efficiency experienced in ozonesondes. The comparison between
165 coincident profile pairs between SAGE III-ISS and the Lauder ozonesonde data was also investigated and shown
in figure 6. A total of 16 coincident pairs were found from June 2017 to May 2018. The coincident pairs had an
average time difference of 23.6 hours, average latitude difference of 1.8°, average longitude difference of 2.0°,
and a 275.5 km average distance. The average percentage difference between SAGE III-ISS and Lauder ozone
concentrations was less than 10% between 15 and 30 km with a low standard error between about 20 and 30 km.

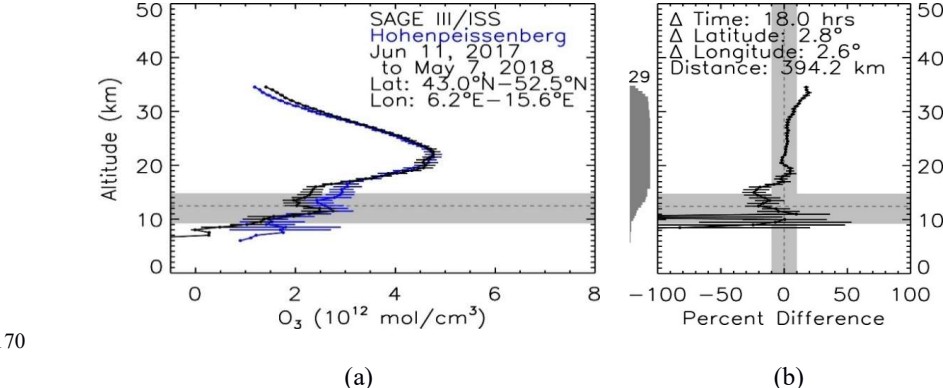

170

(a)                                      (b)

**Figure 5.** Similar to Figure 3 but for the average difference of coincident pairs for SAGE III-ISS and Hohenpeissenburg
ozonesonde profiles obtained for the year June 2017 through May 2018.

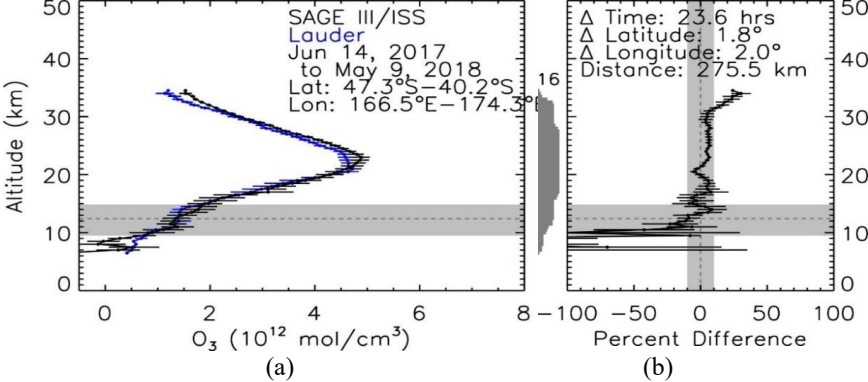

(a)                                      (b)

**Figure 6.** Similar to Figure 3 but for the average difference of coincident pairs for SAGE III-ISS and Lauder ozonesonde
profiles obtained for the year June 2017 through May 2018.



3.3 Comparisons between SAGE III-ISS and ACE-FTS

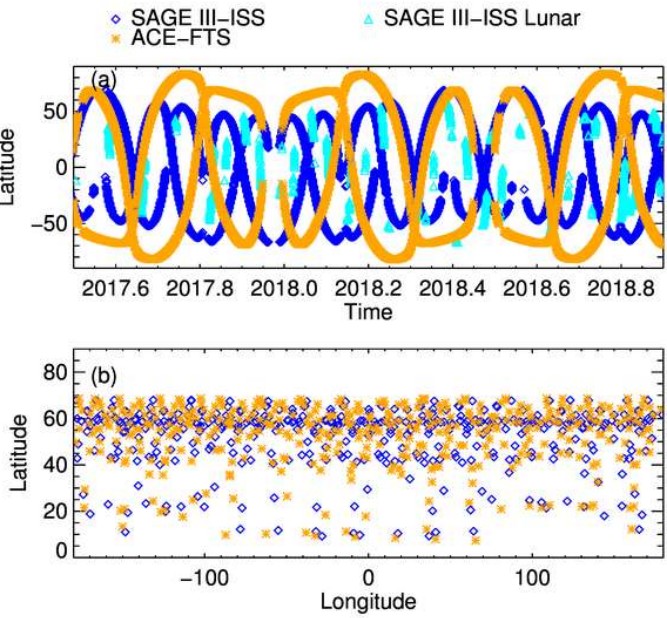

**Figure 7. (a) The ACE-FTS and SAGE III-ISS solar and lunar occultation coverage for comparison. (b) The latitude and longitude distribution for coincident events of SAGE III-ISS and ACE-FTS under the criteria of latitude difference less than ±5°, longitude difference less than ±10°, and time difference less than ±4 hours (only northern hemisphere data shown).**

The coincident ozone concentration between the SAGE III-ISS and ACE-FTS is compared in this section. In order to compare with the ozone mixing ratio data from ACE-FTS, the number density data of SAGE III-ISS was converted to mixing ratio (ppmv) using the temperature and pressure data reported by SAGE III-ISS. The average coincident ozone profile pairs between the SAGE III-ISS and ACE-FTS were compared for different seasons using SAGE III-ISS level 2 solar data from June 2017 to November 2018. Figure 7a shows the ACE-FTS and SAGE III-ISS solar and lunar occultation coverage during the period of comparisons. Figure 7b shows the latitude and longitude distribution of the coincident events between SAGE III-ISS and ACE-FTS with the criteria of latitude difference less than ±5°, longitude difference less than ±10°, and time difference less than ±4 hours. A total of 403 coincident profiles were found using the criteria. Coincident events are mostly located at mid-latitudes and high latitudes in the northern hemisphere because of the high inclination of the SAGE III_ISS and ACE-FTS orbits (Dupuy et al., 2009]. More than 85% of the coincident SAGE III-ISS and ACE-FTS events are located at latitudes higher than 40° N in this case.

Figures 8 a, c, e, and g show the seasonal average ozone mixing ratio profile and their average precision for the two instruments in the Northern Hemisphere for JJA (182 pairs), SON (119 pairs), DJF (39 pairs), and MAM (63 pairs). The SAGE III-ISS average ozone mixing ratio profiles are shown in the blue solid line with the precision shown as horizontal blue error bars. The ACE-FTS average ozone mixing ratio profile is shown as the red solid line with the precision shown as the horizontal red error bar.

Figures 8 b, d, f, and h show the average percentage difference between the two instruments and its standard deviation. The bold black line shows the average percentage difference and the dash-dotted line shows the standard deviation of the percentage difference. The vertical grey regions indicate the ±5% difference region. The mean percentage difference between the two instruments is mostly less than 5% from 20 to 45 km. The





comparisons show slightly larger positive differences near 40 km for the DJF, and MAM measurements. The comparisons show slightly negative differences near 30 km for the SON measurements, and positive differences near 30 km for DJF. Between 20 and 40 km, standard deviations for the percentage differences were less than 5% in JJA and MAM. DJF show the largest standard deviation but still mostly less than 10% between 20 and 50 km. The standard deviation below 20 km is larger for all seasons with a maximum of 50% in DJF.As shown in figure 7b, the coincident events between SAGE III-ISS and ACE-FTS could only be found in northern hemisphere under the criteria of latitude difference less than ±5°, longitude difference less than ±10°, and time difference less than ±4 hours.

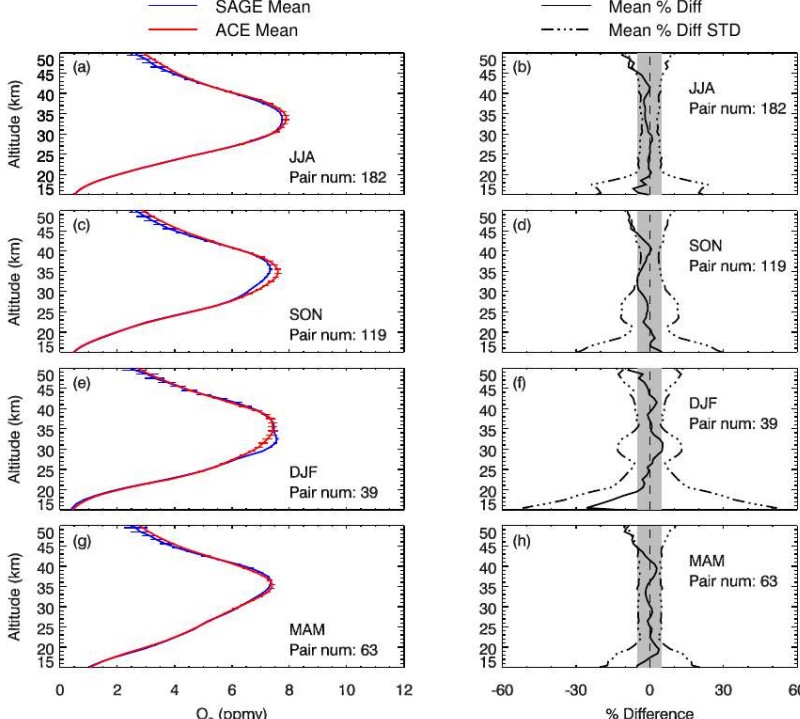

**Figure 8.** Seasonal average ozone mixing ratio profile comparisons between coincident SAGE III-ISS and ACE-FTS measurements for northern hemisphere criteria describe in figure 7.

A wider criteria of latitude difference less than ±10°, longitude difference less than ±20°, and time difference less than ±10 hours was taken to find four season coincident events between the SAGE III-ISS and ACE-FTS in southern hemisphere and a total of 301 pairs of coincident profiles were found. Similar to Figure 8, Figures 9 a, c, e, and g show the seasonal average ozone mixing ratio profile and their average precision for coincident events of the two instruments at the southern hemisphere for JJA (144 pairs), SON (58 pairs), DJF (94 pairs), and MAM (5 pairs).

Figure 9 a, c, e, g show that the mean percentage difference between the two instruments is less than 5 % from 20 to 45 km for SON and DJF. The comparison for JJA show less than 10 % difference between 20 and 50 km with slightly larger positive difference near 30 km and 40 km. The comparison for MAM result a less than 5 % difference between 25 and 45 km.    Between 25 and 45 km, standard deviations for the percentage difference were mostly less than 10 % in SON and DJF. Results in JJA and MAM show the largest standard deviation but



still mostly less than 20 % between 20 and 50 km. The standard deviation below 20 km is larger for all seasons

220    with maximum as large as 60 % in MAM.

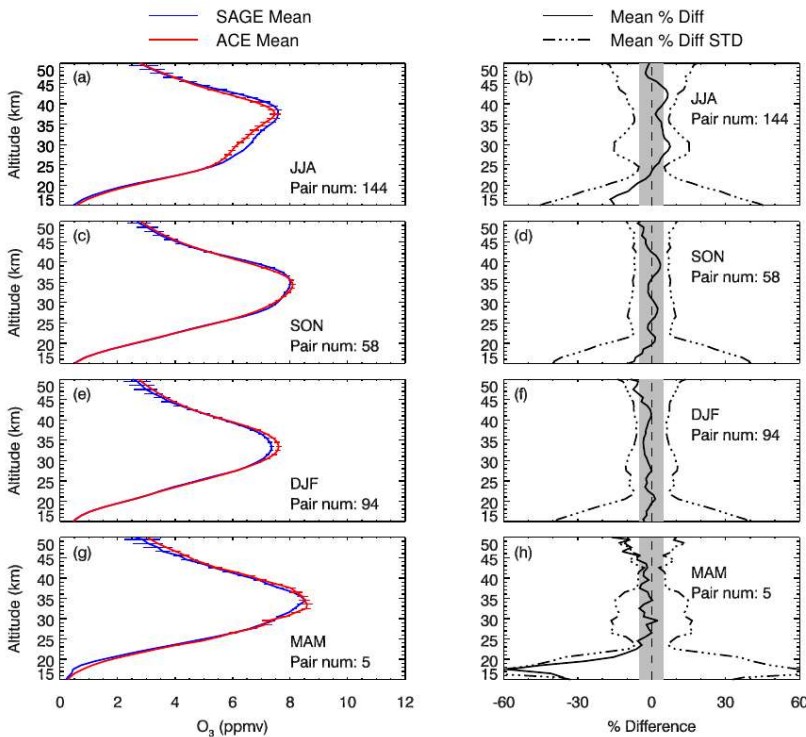

**Figure 9.   Seasonal average ozone mixing ratio profile comparisons between coincident SAGE III-ISS and ACE-FTS measurements for the southern hemisphere under the criteria of differences of latitude less than ±10°, longitude less than ±20°, and time less than ±10 hours.**

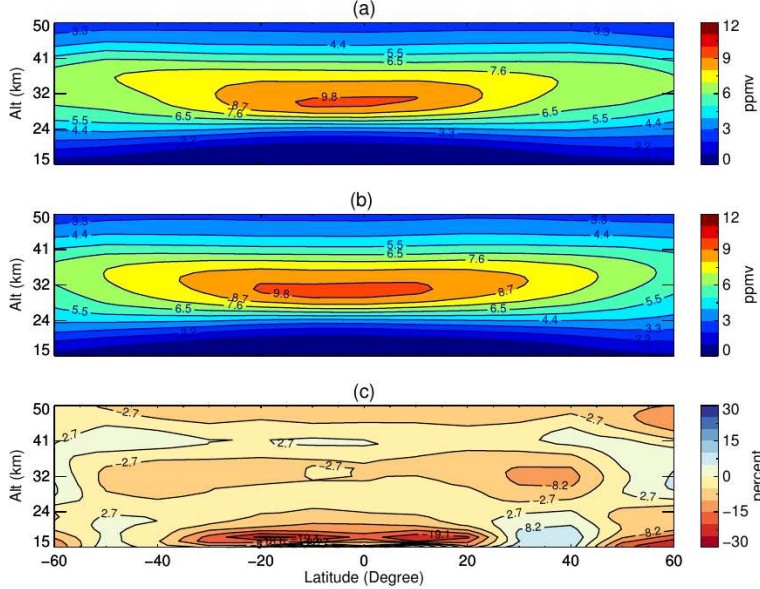

225

**Figure 10. Zonal mean ozone comparisons between SAGE III-ISS and ACE-FTS between 15 and 50 km. (a) The zonal mean ozone for SAGE III-ISS. (b) The zonal mean ozone for ACE-FTS. (c) The zonal mean ozone percentage**
230    **difference between the two instruments.**





The zonal mean ozone mixing ratio from 15 to 50 km for SAGE III-ISS and ACE-FTS were calculated using 10 latitude bins from 60° N to 60° S. Results are shown in figure 10a and b. The maximum ozone mixing ratios located at about 32 km in the tropical region with maximum mixing ratios larger than 9.8 ppmv are shown for both SAGE III-ISS and ACE-FTS. Figure 10c indicates that SAGE III-ISS zonal average ozone mixing ratios are generally less than ACE-FTS throughout the tropical stratosphere with exception near 40 km. The differences are mostly less than 5 % except close to the tropical tropopause area, which is most likely impacted by cirrus clouds.

**3.4 Comparison between SAGE III-ISS Lunar and Lauder ozonesonde profiles**

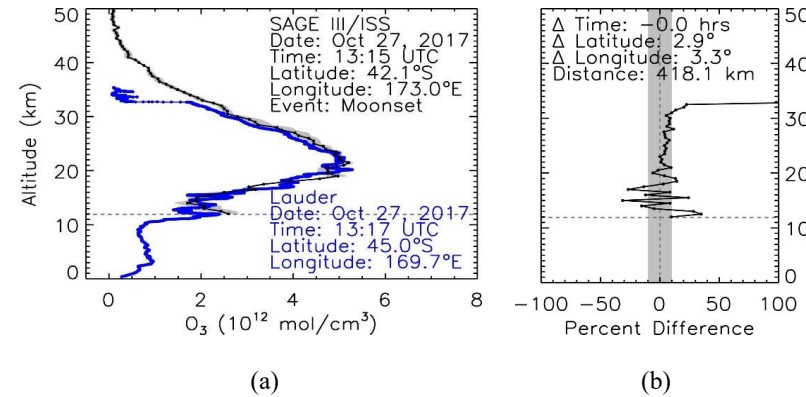

(a)                                        (b)

**Figure 11. Similar to Figure 2, but for a coincident pair for SAGE III-ISS lunar occultation measurements and the Lauder ozonesonde data taken on October 27, 2017.**

The coincident lidar, ozonesonde, and ACE-FTS ozone profiles were compared with SAGE III-ISS Lunar ozone profiles. SAGE III/ISS lunar observations are taken much less frequently than solar observations, and only the lunar data from June 2017 to February, 2019 were available to the authors at the time this analysis was completed. Consequently, only a few coincidences were obtained within the previously defined criteria, including one with a Lauder ozonesonde profile, one with a Lauder lidar measurement, and 7 with ACE-FTS.

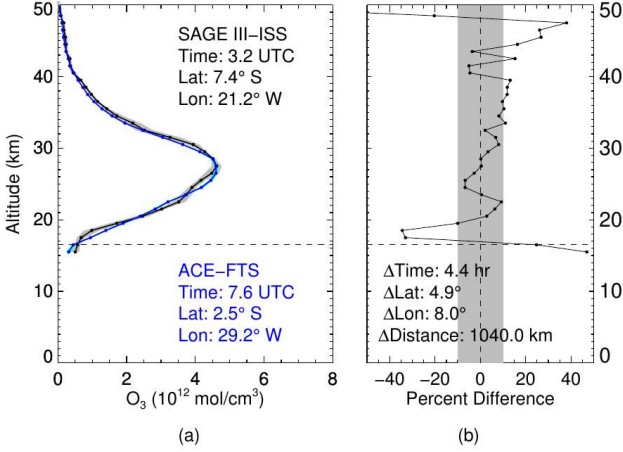

(a)                          (b)

**Figure 12. Similar to Figure 2, but for a coincident pair for SAGE III-ISS lunar occultation measurements and the ACE-FTS ozone profile taken on October 30, 2017.**



Figure 11 shows a coincident event between SAGE III-ISS and Lauder ozonesonde data taken at almost the same time, latitude (difference is 2.9°), longitude (difference is 3.3°), and with a spatial difference of 418.1 km. The percentage difference between the coincident pair is less than 10% between approximately 19 and 31 km. The coincident pair of ozone profiles between SAGE III-ISS and the Lauder lidar (plot not shown) shows similar results of less than a 10% difference between 20 and 30 km. The time difference for this coincident event is 1.3 hours in time, 2.9° in latitude, 3.3° in longitude, and 413.7 km in distance. The coincident pair between SAGE III-ISS lunar and ACE-FTS is shown in Figure 12. The difference between the two coincident pair was less than 10% between 20 and 45 km over most altitudes with a time difference of 4.6 hours, latitude difference of 3.3°, longitude difference of 9.7°, and a separation distance of 1135.9 km.

## 4 Conclusions

This paper represents an early effort to provide validation of ozone to the broad science-user community. It goes a long way to verify the performance of the SAGE III-ISS satellite instrument, and its capability for providing realistic atmospheric ozone profile measurements. The first year of ozone data from SAGE III-ISS was compared with ground-based lidar and ozonesondes. The SAGE III-ISS ozone data were compared with satellite ACE-FTS ozone data from June 2017 to November 2018. More than 700 coincident ozone profile pairs were used for the comparisons and the results show that SAGE III-ISS is capable of accurate ozone profile measurements from the tropopause through the low-to-high stratosphere globally, including during different seasons. Although there are fewer lunar coincidences available, early results suggest that the lunar ozone measurements agree well with ozonesonde and ACE-FTS. It suggests that SAGE III-ISS ozone data are as accurate as a number of correlative measurements and given a reasonable lifetime should be used for stratospheric trend and recovery studies as well as research on the impact of ozone on variation studies. The authors will continue these comparisons into the near future as newer data sets and new versions of SAGE III-ISS level 2 data become available.

*Data availability.* The satellite ozone profile data used in this work were obtain from:
- SAGE III-ISS v5.1 (available at: https://fpd.larc.nasa.gov/sage-iii.html)
-ACE-FTS v3.5/3.6 (available at: https://database.scisat.ca/level2/ace_v3.5_v3.6/)
The ground-based lidar and ozonesonde ozone profile data were obtained from the NDACC Data Host Facility (http://www.ndacc.org)

*Author contribution.* Dr. McCormick formulated the overarching research goals. Dr. Hill compared the ozone profiles from SAGE III-ISS with coincident ozone profiles obtained from lidar and sondes. Lei compared the SAGE III-ISS and ACE-FTS ozone profiles. Lei wrote the initial draft with contributions from all co-authors. Drs. McCormick and Hill reviewed the manuscript and Dr. McCormick provided the final manuscript for submission.

*Competing Interests.* The authors declare that they have no conflict of interest.

*Acknowledgments.* This study is partially supported and monitored by The National Oceanic and Atmospheric Administration – Cooperative Science Center for Earth System Science and Remoting Sensing Technologies (NOAA-CRESSRST) under the Cooperative Agreement Grant #NA16SEC4810008. The statements contained



within the manuscript/research article are not the opinions of the funding agency or the U.S. government, but reflect the author's opinion. We would like to thank the NASA Langley Research Center (NASA-LaRC) Data Archive Center for providing the SAGE III-ISS solar and lunar occultation data. The ground-based lidar and ozonesonde data used in this publication were obtained from the Hohenpeissenberg Meteorological Observatory,

German National Meteorological Hohenpeissenberg, Germany, and the National Institute of Water and Atmospheric Research (NIWA), Lauder, Omakau, Central Otago, New Zealand, as part of the Network for the Detection of Atmospheric Composition Change (NDACC). The Atmospheric Chemistry Experiment (ACE), also known as SCISAT, is a Canadian-led mission mainly supported by the Canadian Space Agency. We want to thank the ACE team for providing the ACE-FTS ozone data used in this work.

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
