# Peer review of "Early Results and Validation of SAGE III-ISS Ozone Profile Measurements from Onboard the International Space Station"

_Atmospheric Measurement Techniques, 2019_

## Referee Comment (RC1) · Anonymous Referee #1 · 7 Nov 2019

General comments:

This work provides the first validation results of the SAGE III-ISS ozone profile observations with respect to ozonesonde, lidar, and ACE-FTS measurements. Although results are usually presented in a clear way, the validation methodology and its description (mainly Section 2.4, see below) should be improved before publication in AMT.

Specific comments:

- Abstract, line 20 (and throughout the text): The meaning of the term 'accuracy' is unclear as it is sometimes differently used in different contexts. In fact, 'accuracy' should not be used as a synonym of total uncertainty; see VIM, GUM, or Loew et al.

[Figure]

(2017) for their application to satellite validation practices. It is recommended to just use 'total uncertainty' instead.

- Lines 66-70: The 'CompositeO3' product is not explained in the last paragraph of Section 2.1.

- Section 2.2: Information on the ozonesonde launch frequency and lidar observation frequency is missing, while this is of importance in considering the reference data coverage (temporal representativeness). Typical uncertainties of these measurements, with references to the appropriate literature, should be mentioned as well for proper interpretation of the comparison statistics later on.

- Section 2.4 is rather poorly written in terms of English language quality (e.g. lines 91-92), contains duplicate information, and the comparison statistics can be significantly improved. Regarding the latter, three important issues have been detected: 1. The coincidence criteria are not motivated, except for the mentioning of the need for larger values to compensate for the fewer comparisons in the southern hemisphere. The coincidence criteria should be motivated, ideally in terms of the (estimated) spatial extent of the measurements, including references. 2. Using a linear interpolation (line 105) for vertical sampling to a common grid is presently considered to leave important vertical profile information out of the resampling. Improved methods given in Calisesi et al. (2005) or Langerock et al. (2015) (or see Keppens et al. (2019) for an overview) could be implemented. 3. Calculating relative differences by using the averaged observation as a reference (lines 115-117) is not appropriate for satellite validation exercises (it is rather used for model comparisons as the average model is typically closer to the truth than each individual model): The reference averaging process corrupts the independency of the reference and smooths the effects that you want to detect (the unknown satellite errors), making their quantification flawed. If one wants to normalize the satellite deviation, the normalization should be done independently from the satellite data, i.e. using the reference data only.

- Section 3: In order to properly address the satellite uncertainty budget, the spatiotemporal sampling difference errors and vertical smoothing difference errors that result from the inexact coincidences should be discussed as part of the uncertainties. E.g. one expects an effect of the larger coincidence criteria in the southern hemisphere.

- Line 237: Please provide a reference for the statement that the tropical tropopause area "is most likely impacted by cirrus clouds".

- The conclusions should contain quantitative results (as the abstract).

Technical corrections:

- Abstract, line 14: "the average difference of ozone concentration measured by SAGE III-ISS" is a confusing phrasing (as if average differences are measured). Please rephrase.

- Line 43: Explain the acronym 'CFC'.

- Line 103: Provide links or references to the product user guides.

- Line 190: Explain three-monthly acronyms.

- Conclusions, line 258: "This paper represents an early effort to provide validation of ozone to the broad science-user community." is a too broad statement. Which ozone, and measured by what?

---

## Referee Comment (RC2) · Anonymous Referee #2 · 8 Nov 2019

**Short resume**

This paper discusses the first validation activities on SAGE III-ISS ozone occultation measurements. These data continue and add value to the long-term SAGE time series. As a reference for the validation, two stations equipped with ground-based lidar and ozonesonde are selected. In addition, a comparison is performed with respect to ACE-FTS observations. The early results of the validation are presented in a structured way with a clear description of the procedure followed for the comparison. The description of the results could be improved.

[Figure]

This paper fits the scope of AMT, and it is logically written. From my side, I have some comments on specific aspects and technical corrections.

**Specific comments**

1. Usage of SAGE III-ISS profiles
   In section 2.1, the authors introduce the four retrieved ozone products but I could not find in the text which one is then used for the data analysis. Could you please provide some more details about this point and, if available, some characterization of the retrieved profiles? For example, the vertical resolution of ACE-FTS ozone profiles is mentioned in Section 2.3 and a similar information for SAGE III-ISS would be interesting as well. Could you add a reference about the retrieval algorithm, if available, and the url of the Data Products User's Guide at line 70?

2. Why were precisely Lauder and Hohenpeissenberg stations selected instead of several stations at northern and southern mid-latitudes? Is there any specific reason?

3. If I understood it correctly, the authors have always used a linear interpolation to compare higher resolution profiles from correlative data sets with SAGE III-ISS observations. Did you think about the use of averaging kernels from SAGE III-ISS retrievals, especially when comparing ozonesonde data? Their use may improve the comparison in the lower stratosphere and reduce oscillations in the relative differences (e.g. Fig. 11).

4. Collocation criteria
   I found the collocation criteria used for ozonesonde/lidar particularly loose in terms of time coincidence (a window of 96 hours around the observations, right?) and pretty strict in the spatial domain: the longitude requirement of 5° could be in
my mind slightly relaxed to find more collocations, as chosen for ACE-FTS. Could you justify this choice of criteria? A general better motivation of the collocation criteria would be appreciated.

5. I found the description of the second part of Sect. 2.4 clear and well structured. On the contrary, I found confusing the quantities shown in Figs. 3 and following. In the left panel, do the error bars correspond to the standard deviation, like the quantity in Eq. (3), or is it the average of the precision of the considered profiles (which, however, in Fig. 2 was almost not visible)? In the right panel, do you plot the percentage difference between the averaged profiles from the two instruments (as I understood from the caption) or $\Delta(z)$ from Eq.(1)? Is the standard error equal to the SEM of Eq. (4)?

6. The use of $ref_i(z)$ as described in Eq. (2) is to me well justifiable for the comparison with ACE-FTS profiles, being also satellite observations, which need to be validated against ground-based data. On the contrary, when comparing SAGE III-ISS with ozonesonde and lidar profiles I would use the reference profile itself in the denominator instead of Eq. (2). This is also done by Randall et al. (2003), which is also cited by the authors.

7. I am confused with the collocations between SAGE III-ISS and ACE-FTS shown in Fig. 7b. From the beginning it is to me not clear that these are the only available collocations. On the contrary, it seems that the plot refers to the northern hemisphere only for graphical purposes, as Fig. 7 caption says 'only northern hemisphere data shown'. Then at line 185: 'Coincident events are MOSTLY located at mid-latitudes and high-latitudes in the northern hemisphere...'. On the next page, I find the clear statement at line 203: '...coincident events [...] could only be found in the northern hemisphere under the criteria...'. Could you please clarify this and, in case, move this last sentence up to line 185?

8. I find panels (a) and (b) of Fig. 10 not so quantitative interesting, as it is difficult

to estimate the differences between the two plots by eye. I would rather expand panel (c) of this picture and leave it alone so that the differences as a function of latitude and altitude are better visible.

9. At line 237, cirrus clouds are indicated as responsible for discrepancies between SAGE III-ISS and ACE-FTS, how do you filter out clouds from measurements?

10. For the lunar occultation, why don't you show the average of the 7 available coincidences with ACE-FTS instead of one example only (Fig. 12)?

11. The layout of the figures
I think that the layout and visibility of some pictures would strongly improve by reducing the font of the writings in the plots. I consider Fig. 2 as an example but this is applicable also to Figs. 3, 4, 5, 6, 11 and 12. On the left panel, could you please reduce the text information in the plot area and their font size? In this way, the x-axis could be zoomed to the range 0 - 6 $* 10^{12}$, and the profiles would be more visible. For example, the information about latitude and longitude of the station is constant and is already mentioned in Sect. 2.2. I also find the x-axis of the right panel too wide: it would be more interesting to better see the differences in the 15-40 km range. I suggest expanding the range to [-50, 50]. Indeed, the large relative differences in the lowermost and uppermost layers are anyway cut in some plots, even keeping the [-100, 100] range. In addition, the text on the right panel is, in my opinion, redundant, as always mentioned in the text as well. By removing this, you avoid the overlap of some writing with the profiles, as in Fig. 4. Similar suggestions, though not so critical as before, I would recommend for Figs. 8 and 9. Expanding the x-axis would ease the visibility of the profiles and of the differences.

12. Verb Tenses
I noted that in different parts of the paper inconsistent verb tenses are used. For example, in Sect. 2.4, after using the past tense to describe the collocation

procedure, the future/present tense is introduced. Another example: in Sect. 3.2, the whole description of Figs. 5 and 6 is in the past, whereas Fig. 4 is described in the present.

**Technical corrections**

P1, l28: I would also mention the impact of the climate change on the ozone layer (e.g. BDC modification, stratospheric cooling), not only the impact of ozone changes in the climate.

P1, l41: 'record of observation' → 'record of observations'

P2, l46: change to 'SAGE III-ISS ozone profiles with observations made by ...'

P4, l124-126: I would avoid repeating the description of the single terms, which are already explained after Eq. (1).

P4, l131: As the last paragraph title includes the lunar adjective, I would also specify here 'SAGE III-ISS solar' measurements.

P4, caption Fig. 2: check the spelling of Hohenpeissenberg

P5, caption Fig. 3: check the spelling of Hohenpeissenberg

P5, l145: I would delete one ozone in the expression 'Hohenpeissenberg ozone lidar ozone data'

P6, l158: Same as for the title of Sect. 3.1

P6, l164: please add a reference to the statement about the pump efficiency.

P7, top: something is wrong with the font of the section title.

P7, l187: wrong parenthesis, ] → )

P7, l189 and l211: profile → profiles

P7, l191-206: I think it is not necessary to repeat the legend of the figure in the text.

P8, l207: describe → described

P8, l215: 'Figure 9 a, c, e, g' replace with 'Figures 9 b, d, f, h'

P8, l216: result → shows

P11, l252-253: I would delete the sentence 'The time difference for this ... in distance'.
P11, l268: what does it mean 'the impact of ozone on variation studies'?
P11, l272: were obtain → were obtained.
The term 'northern hemisphere' is written sometimes capitalized and other times small.
Please check the correct use of Fig. and Figs., e.g lines 210, 214, 233.

---

## Author Comment (AC1) · 8 Jan 2020

General comments: This work provides the first validation results of the SAGE III-ISS ozone profile observations with respect to ozonesonde, lidar, and ACE-FTS measurements. Although results are usually presented in a clear way, the validation methodology and its description (mainly Section 2.4, see below) should be improved before publication in AMT.

We want to thank the referee for taking the time to review our paper and provide us with valuable comments and suggestions for its improvement.

Specific reviewer comments and author responses:

1. Abstract, line 20 (and throughout the text): The meaning of the term 'accuracy' is unclear as it is sometimes differently used in different contexts. In fact, 'accuracy' should not be used as a synonym of total uncertainty; see VIM, GUM, or Loew et al. (2017) for their application to satellite validation practices. It is recommended to just use 'total uncertainty' instead.

Response: We agree with the reviewer and have changed the text to reflect this throughout the revised manuscript.

2. Lines 66-70: The 'CompositeO3' product is not explained in the last paragraph of Section 2.1.

Response: The 'CompositeO3' product, which was included in v5.0 of the SAGE III-ISS data, was an attempt by the SAGE III team to combine the other three ozone profile products into a single profile. This product was removed from v5.1 of the SAGE III-ISS data and is not used in the comparisons, therefore, the statement that mentions the 'CompositeO3' profile has been deleted in the revised manuscript.

3. Section 2.2: Information on the ozonesonde launch frequency and lidar observation frequency is missing, while this is of importance in considering the reference data coverage (temporal representativeness). Typical uncertainties of these measurements, with references to the appropriate literature, should be mentioned as well for proper interpretation of the comparison statistics later on.

Response: Information on the ozonesonde launch frequency and lidar observation frequency for the June 2017 to May 2018 time period, as well as typical uncertainties

of the measurements with references, have been added in two extra paragraphs at the end of Section 2.2.

4. Section 2.4 is rather poorly written in terms of English language quality (e.g. lines 91- 92), contains duplicate information, and the comparison statistics can be significantly improved. Regarding the latter, three important issues have been detected: 1. The coincidence criteria are not motivated, except for the mentioning of the need for larger values to compensate for the fewer comparisons in the southern hemisphere. The coincidence criteria should be motivated, ideally in terms of the (estimated) spatial extent of the measurements, including references. 2. Using a linear interpolation (line 105) for vertical sampling to a common grid is presently considered to leave important vertical profile information out of the resampling. Improved methods given in Calisesi et al. (2005) or Langerock et al. (2015) (or see Keppens et al. (2019) for an overview) could be implemented. 3. Calculating relative differences by using the averaged observation as a reference (lines 115-117) is not appropriate for satellite validation exercises (it is rather used for model comparisons as the average model is typically closer to the truth than each individual model): The reference averaging process corrupts the independency of the reference and smooths the effects that you want to detect (the unknown satellite errors), making their quantification flawed. If one wants to normalize the satellite deviation, the normalization should be done independently from the satellite data, i.e. using the reference data only.

Response: Section 2.4 has been rewritten to improve the grammar and clarify the coincidence criteria. -1. A reference has been added to explain the motivation for selecting coincident measurements. -2. Although beyond the scope of this initial investigation, other techniques of resampling to a common grid, including the use of averaging kernels, will be considered for future validation efforts. -3. The ozonesonde/lidar comparisons have been recomputed using the ozonesonde/lidar data as the reference in the denominator (instead of the average between the instruments). A statement was added to Section 2.4 after Eq. (2) to clarify this point. The ozonesonde/lidar comparison figures (3-6 and 11) and associated text were modified according to the new results.

5. Section 3: In order to properly address the satellite uncertainty budget, the spatiotemporal sampling difference errors and vertical smoothing difference errors that result from the inexact coincidences should be discussed as part of the uncertainties. E.g. one expects an effect of the larger coincidence criteria in the southern hemisphere.

Response: We agree with the reviewer and added a statement indicating that larger errors are expected using an expanded coincidence criteria. A sentence at the end of 2.4 has been added. A more detailed quantitative analyses would require a larger data record than available in this early validation manuscript.

6. Line 237: Please provide a reference for the statement that the tropical tropopause area "is most likely impacted by cirrus clouds".

Response: A reference was added to support the statement that cirrus clouds occur more frequently in the tropical tropopause region: Nazaryan et al., 2008.

7. The conclusions should contain quantitative results (as the abstract).

Response: Several sentences have been added to the conclusions that supply quantitative results similar to what was presented in the abstract.

Technical corrections: 8. Abstract, line 14: "the average difference of ozone concentration measured by SAGE III-ISS" is a confusing phrasing (as if average differences are measured). Please rephrase.

Response: The sentence in the abstract has been replaced with "Average differences in ozone concentration between SAGE III-ISS and Hohenpeissenberg lidar observations for one year are less than 10% between 16 and 42 km and less than 5% between 20 and 40 km."

9. Line 43: Explain the acronym 'CFC'.

Response: The acronym 'CFC' has been replaced with "Chlorofluorocarbons (CFC)" in the text.

10. Line 103: Provide links or references to the product user guides.

Response: A link to the SAGE III-ISS Data Products User's Guide has been added.

11. Line 190: Explain three-monthly acronyms.

Response: Parenthetical text has been added to explain the three-monthly acronyms.

12. Conclusions, line 258: "This paper represents an early effort to provide validation of ozone to the broad science-user community." is a too broad statement. Which ozone, and measured by what?

Response: For clarification, the sentence has been replaced by "This paper represents an early effort to provide validation of upper tropospheric and stratospheric ozone measurements from SAGE III-ISS to the broad science-user community".
* * *

---

## Author Comment (AC2) · 8 Jan 2020

Short resume This paper discusses the first validation activities on SAGE III-ISS ozone occultation measurements. These data continue and add value to the long-term SAGE time series. As a reference for the validation, two stations equipped with ground-based

lidar and ozonesonde are selected. In addition, a comparison is performed with re-spect to ACE-FTS observations. The early results of the validation are presented in a structured way with a clear description of the procedure followed for the comparison. The description of the results could be improved. This paper fits the scope of AMT, and it is logically written. From my side, I have some comments on specific aspects and technical corrections.

We want to thank the referee for taking the time to review our paper and provide us with valuable comments and suggestions for its improvement.

Specific Reviewer comments and author responses

1. Usage of SAGE III-ISS profiles In section 2.1, the authors introduce the four retrieved ozone products but I could not find in the text which one is then used for the data analysis. Could you please provide some more details about this point and, if available, some characterization of the retrieved profiles? For example, the vertical resolution of ACE-FTS ozone profiles is mentioned in Section 2.3 and a similar information for SAGE IIIISS would be interesting as well. Could you add a reference about the retrieval algorithm, if available, and the url of the Data Products User's Guide at line 70?

Response: The vertical resolution of the SAGE III-ISS ozone profile is 0.5 km altitude. The version 5.1 Least Squares Ozone profiles are used for the comparisons with the correlative measurements. The detailed description of the retrieval algorithm can be found in Damadeo et al. (2013). The url for the SAGE III-ISS Data Product Users Guide is: https://eosweb.larc.nasa.gov/project/sageiii-iss/guide/DPUG-G3B-2-0.pdf. This text has been incorporated in the revised manuscript.

2. Why were precisely Lauder and Hohenpeissenberg stations selected instead of several stations at northern and southern mid-latitudes? Is there any specific reason?

Response: An NDACC station at mid-latitude in each hemisphere with an established record of regular ozone measurements from both lidars and sondes was selected for

the initial validation. There is no other specific reason for these two particular stations. Future validation efforts will include other stations. This information was added at the beginning of section 2.2 in the revised manuscript.

3. If I understood it correctly, the authors have always used a linear interpolation to compare higher resolution profiles from correlative data sets with SAGE III-ISS observations. Did you think about the use of averaging kernels from SAGE III-ISS retrievals, especially when comparing ozonesonde data? Their use may improve the comparison in the lower stratosphere and reduce oscillations in the relative differences (e.g. Fig. 11).

Response: The use of averaging kernels is beyond the scope of this initial work, but will be considered for future validation efforts.

4. Collocation criteria I found the collocation criteria used for ozonesonde/lidar particularly loose in terms of time coincidence (a window of 96 hours around the observations, right?) and pretty strict in the spatial domain: the longitude requirement of 5_ could be in my mind slightly relaxed to find more collocations, as chosen for ACE-FTS. Could you justify this choice of criteria? A general better motivation of the collocation criteria would be appreciated.

Response: Collocation criteria for the ozonesonde/lidar comparisons were modified to reduce the time range to 24 hours and expand the longitude range to $10°$. One result of this modification was that observations from one instrument were occasionally coincident with multiple observations from another instrument. To avoid comparisons with duplicate measurements, first the SAGE III-ISS profiles were examined for redundancy, and only the ground measurement that was closest in time to the SAGE observation was retained. Then, the ground-based profiles were examined for redundancy, and only the SAGE profile that was closest in distance to the ground observation was retained. Also, as a result of the new criteria Figure 2 and associated text were removed from the paper (since the time separation between the observations was >24 hours),

and the other ozonesonde/lidar comparison figures (3-6 and 11) and text were modified according to the new results.

5. I found the description of the second part of Sect. 2.4 clear and well structured. On the contrary, I found confusing the quantities shown in Figs. 3 and following. In the left panel, do the error bars correspond to the standard deviation, like the quantity in Eq. (3), or is it the average of the precision of the considered profiles (which, however, in Fig. 2 was almost not visible)? In the right panel, do you plot the percentage difference between the averaged profiles from the two instruments (as I understood from the caption) or _(z) from Eq.(1)? Is the standard error equal to the SEM of Eq. (4)?

Response: The error bars in both panels of Fig. 3 show twice the standard error of the respective means [2 $\times$ Eq. (4)]. The right panel shows the average percent difference of the comparisons [Eq. (1)]. The caption and text have been modified to make this clearer.

6. The use of refi(z) as described in Eq. (2) is to me well justifiable for the comparison with ACE-FTS profiles, being also satellite observations, which need to be validated against ground-based data. On the contrary, when comparing SAGE III-ISS with ozonesonde and lidar profiles I would use the reference profile itself in the denominator instead of Eq. (2). This is also done by Randall et al. (2003), which is also cited by the authors.

Response: The ozonesonde/lidar comparisons have been recomputed using the ozonesonde/lidar data as the reference in the denominator. The ozonesonde/lidar comparison figures (3-6 and 11) and associated text were modified according to the new results.

7. I am confused with the collocations between SAGE III-ISS and ACE-FTS shown in Fig. 7b. From the beginning it is to me not clear that these are the only available collocations. On the contrary, it seems that the plot refers to the northern hemisphere only for graphical purposes, as Fig. 7 caption says 'only northern hemisphere data

shown'. Then at line 185: 'Coincident events are MOSTLY located at mid-latitudes and high-latitudes in the northern hemisphere...'. On the next page, I find the clear statement at line 203: '...coincident events [...] could only be found in the northern hemisphere under the criteria...'. Could you please clarify this and, in case, move this last sentence up to line 185?

Response: Only Northern Hemisphere coincident events were found with the defined criteria. The associated text was modified in the revised manuscript (see figure caption) as suggested by the reviewer.

8. I find panels (a) and (b) of Fig. 10 not so quantitative interesting, as it is difficult to estimate the differences between the two plots by eye. I would rather expand panel (c) of this picture and leave it alone so that the differences as a function of latitude and altitude are better visible.

Response: We agree with the reviewer and have expanded figure 10c in the revised manuscript.

9. At line 237, cirrus clouds are indicated as responsible for discrepancies between SAGE III-ISS and ACE-FTS, how do you filter out clouds from measurements?

Response: We do not specifically screen for clouds. We screen the SAGE III-ISS and ACE-FTS data based on the data quality flags determined by the respective instrument teams. The text reflecting this has been added in the revised manuscript.

10. For the lunar occultation, why don't you show the average of the 7 available coincidences with ACE-FTS instead of one example only (Fig. 12)?

Response: We have added text and a figure with respect to the average of the 7 available coincidences and associated statistics in the revised manuscript as suggested by the reviewer.

11. The layout of the figures I think that the layout and visibility of some pictures would strongly improve by reducing the font of the writings in the plots. I consider Fig. 2

as an example but this is applicable also to Figs. 3, 4, 5, 6, 11 and 12. On the left panel, could you please reduce the text information in the plot area and their font size? In this way, the x-axis could be zoomed to the range 0 - 6 _ 1012, and the profiles would be more visible. For example, the information about latitude and longitude of the station is constant and is already mentioned in Sect. 2.2. I also find the x-axis of the right panel too wide: it would be more interesting to better see the differences in the 15-40 km range. I suggest expanding the range to [-50, 50]. Indeed, the large relative differences in the lowermost and uppermost layers are anyway cut in some plots, even keeping the [-100, 100] range. In addition, the text on the right panel is, in my opinion, redundant, as always mentioned in the text as well. By removing this, you avoid the overlap of some writing with the profiles, as in Fig. 4. Similar suggestions, though not so critical as before, I would recommend for Figs. 8 and 9. Expanding the x-axis would ease the visibility of the profiles and of the differences.

Response: The text has been removed from Figs. 3-6 and 11, and the information has been placed in the captions and narrative. The range of the x-axis in the left panels has been changed to 0 to $6\times1012$, and the range of the x-axis in the right panels has been changed to -50 to 50.

12. Verb Tenses I noted that in different parts of the paper inconsistent verb tenses are used. For example, in Sect. 2.4, after using the past tense to describe the collocation procedure, the future/present tense is introduced. Another example: in Sect. 3.2, the whole description of Figs. 5 and 6 is in the past, whereas Fig. 4 is described in the present.

Response: Verb tenses have been changed throughout the revised manuscript to be more consistent.

Technical corrections 13. P1, l28: I would also mention the impact of the climate change on the ozone layer (e.g. BDC modification, stratospheric cooling), not only the impact of ozone changes in the climate.

Response: A statement on the impact of climate change on stratospheric ozone concentrations has been added in the revised manuscript with its reference.

14. P1, l41: 'record of observation' ! 'record of observations'

Response: This has been changed in the revised manuscript.

15. P2, l46: change to 'SAGE III-ISS ozone profiles with observations made by ...'

Response: This has been changed in the revised manuscript.

16. P4, l124-126: I would avoid repeating the description of the single terms, which are already explained after Eq. (1).

Response: This has been changed in the revised manuscript.

17. P4, l131: As the last paragraph title includes the lunar adjective, I would also specify here 'SAGE III-ISS solar' measurements.

Response: This has been changed in the revised manuscript.

18. P4, caption Fig. 2: check the spelling of Hohenpeissenberg

Response: This has been corrected in the revised manuscript.

19. P5, caption Fig. 3: check the spelling of Hohenpeissenberg

Response: This has been corrected in the revised manuscript.

20. P5, l145: I would delete one ozone in the expression 'Hohenpeissenberg ozone lidar ozone data'

Response: The expression was changed to "Hohenpeissenberg lidar ozone data".

21. P6, l158: Same as for the title of Sect. 3.1

Response: This has been changed in the revised manuscript.

22. P6, l164: please add a reference to the statement about the pump efficiency.
Response: The statement about pump efficiency has been replaced with "where uncertainties in the Brewer-Mast ozonesonde measurements rapidly increase (Kerr et al. 1994)", which includes the reference given.

23. P7, top: something is wrong with the font of the section title.

Response: This has been corrected in the revised manuscript.

24. P7, l187: wrong parenthesis, ] ! )

Response: This has been corrected in the revised manuscript.

25. P7, l189 and l211: profile ! profiles

Response: This has been corrected in the revised manuscript.

26. P7, l191-206: I think it is not necessary to repeat the legend of the figure in the text.

Response: This has been changed in the revised manuscript.

27. P8, l207: describe ! described

Response: This has been corrected in the revised manuscript.

28. P8, l215: 'Figure 9 a, c, e, g' replace with 'Figures 9 b, d, f, h'

Response: This has been corrected in the revised manuscript.

29. P8, l216: result ! shows

Response: This has been changed in the revised manuscript.

30. P11, l252-253: I would delete the sentence 'The time difference for this ... in distance'.

Response: This has been changed in the revised manuscript.

31. P11, l268: what does it mean 'the impact of ozone on variation studies'?

Response: It was meant to say "the impact of ozone on climate variation studies". This has been changed in the revised manuscript.

32. P11, l272: were obtain ! were obtained.

Response: This has been changed in the revised manuscript.

33. The term 'northern hemisphere' is written sometimes capitalized and other times small. Please check the correct use of Fig. and Figs., e.g lines 210, 214, 233.

Response: This has been changed in the revised manuscript.

---

## Referee Report (RR1)

I thank the authors for the revision of the manuscript, which improved the readability of the text and of the figures. The authors replied satisfactorily to most of my previous suggestions. I just have few more technical comments.

**Technical corrections**

P4, l132: What does the reference to Hubert et al. 2016 refer to? Is a reference for that sentence needed at all?

P5, l151: Considering the term $\delta_i = \frac{SAGEIII_i(z) - corr_i(z)}{ref_i(z)}$ in Eq. 1, it seems to me, if I am not wrong, that the same term $\delta$ should appear in Eq. 3, which means: $\sum_{i=1}^{N(z)} [(\frac{SAGEIII_i(z) - corr_i(z)}{ref_i(z)} - \Delta(z))]^2$, in order to compute the standard deviation of the distribution of the relative differences.

P6, l178: Could you move the sentence about the vertical spread of the comparisons to Fig. 2, which is the first plot where this vertical gray bar appear?

P8, 197: The coincident ozone concentration between the SAGE III-ISS and ... $\rightarrow$ The coincident ozone concentration between SAGE III-ISS and ...

P8, l202-203: I would rewrite the sentence as: "Coincident events are located in the Northern Hemisphere mostly at mid- and high-latitudes ...'

P8, l206 : Referring to the error bars in Fig. 7 panels a, c, e, g: I don't understand the term 'standard deviation of their uncertainties'. Or better, why should you plot the std of the uncertainties of the single profiles? Didn't you plot the standard error of the profiles as done in the previous figures?

P8, l209-212: I would move these 2 sentences to the caption of Fig. 7.

P9, l222-224: I would also move this sentence at the beginning of the section 3.3, as it refers to Fig. 6d and collocations, which are introduced at the beginning of the section.

P12, Fig.11: The vertical gray bar is here missing.

---

## Author Response (AR2)

Response to Reviewer of proposed AMT paper entitled: Early Results and Validation of SAGE III-ISS Ozone Profile Measurements from Onboard the International Space Station

McCormick et al.

All suggestions made by the reviewer were incorporated into the revised manuscript.

(1) We removed the reference to Hubert et al., 2016, from both the text and reference list as not being needed
(2) Equation 3 was corrected
(3) Sentence regarding vertical grey bar spread of the comparisons was moved to the previous paragraph closer to Fig 2
(4) The sentences in lines 197 and 202-203 were reworded
(5) The error bars in Figs 7 and 8, panels a, c, e and g, were changed to twice the standard error of the mean to make them consistent with the other plots, and the phrase "standard deviation of their uncertainties" was changed accordingly
(6) The sentences in lines 209 to 212, and 222 to 224 were moved
(7) A vertical grey bar showing the spread of comparisons was added to Fig 11, and error bars in panel (a) were recalculated to reflect twice the standard error to be consistent with the other plots